# The Displacement of Santiago de Chile's Downtown during 1990–2015: Travel Time Effects on Eradicated Population

**Gonzalo Suazo-Vecino [1],\*** , **Juan Carlos Muñoz [1]** and **Luis Fuentes Arce [2]**

1   Department of Transport Engineering and Logistics and Centro de Desarrollo Urbano Sustentable (CEDEUS), Pontificia Universidad Católica de Chile, Santiago 6904411, Región Metropolitana, Chile; jcm@ing.puc.cl
2   Institute of Urban and Territorials Studies and Centro de Desarrollo Urbano Sustentable (CEDEUS), Pontificia Universidad Católica de Chile, Santiago 6904411, Región Metropolitana, Chile; lfuentes@uc.cl
\*   Correspondence: gsuazovecino@uc.cl

**Abstract:** The center of activities of Santiago de Chile has been continuously evolving towards the eastern part of the city, where the most affluent residents live. This paper characterizes the direction and magnitude of this evolution through an indicator stating how much the built surface area for service purposes grows in different areas in the city. To identify the impact of this evolution, we compare residents' travel-time distributions from different sectors in the city to the central area. This travel-time comparison is focused on the sectors where informal settlements were massively eradicated between 1978–1985 and those areas where the settlements were relocated. This analysis show that this policy and the consequent evolution of the city were detrimental to the affected families, significantly increasing average travel time to the extended center of the city and inequality among different socioeconomic groups in the city. Although the phenomenon is quite visible to everyone, it has not received any policy reaction from the authority. These findings suggest that middle and low-income sectors would benefit if policies driving the evolution of the center of activities towards them were implemented.

**Keywords:** urban structure; travel times; eradicated population; informal settlements

## 1. Introduction

Many Latin American countries are among those with the highest income inequality worldwide. In many cities in this region, this inequality takes a spatial dimension by showing strong urban segregation and vast differences in the quality of the built environment in high- and low-income areas. In many of these cities, social and informal housing is installed where the land is cheaper and no conflicts with nearby residents will be created (i.e., in periphery areas, away from the most vibrant sectors in the city, often expanding the city borders). Many of these countries also have quite loose land use regulation, allowing real estate builders to locate new developments according to market opportunities.

This type of urban expansion hides a worrying phenomenon: the decisions about where to locate new job opportunities are taken by owners or managers of these new offices. When urban segregation is very high, it appears that these decision makers will most likely choose locations they perceive as safe, where the quality of the urban environment is high, and ideally close to their own residences. If this happens, urban development will become a tool that creates inequality through not only increasing the gap in the opportunities found in different income neighborhoods, but also by increasing the land value difference between these areas. It will also increase the commuting travel times of low- and middle-income workers, while decrease those of high-income workers.

The goal of this paper is to add evidence to this worrying phenomenon that is quite likely related to the significant riots that happened in October and November 2019 in several Chilean and Colombian cities. An important dimension of the complaints of residents in these cities is the long commute time to reach their jobs. In the case of Santiago de Chile, the riots started because of fare evasion, with rioters torching several metro stations and multiple buses. In Colombia, the discontent also focused on their transport system.

This unequal access to services has been captured by several attempts to model Latin American cities [1–3]. For example, the case study of Montevideo carried out by Hernández and Rossel [4] is interesting because it explores "the role of space–time constraints in determining the conditions under which people access basic social services." In it, the empirical evidence reflects that the uncertainty related to the time spent using transport services may have a distorting effect on household's activities.

In this work, we analyze the case of Santiago, studying the relationship between the consolidation of its metropolitan centrality towards the affluent northeastern sector of the city and its impact on travel times for inhabitants of different areas of the city. We focus the travel time comparison on a group of informal settlers who were relocated in Santiago between 1978 and 1995.

With this, we hope to identify an important source of urban inequality and characterize its worrying trend. This source impacts the quality of life experienced by low income sectors, widening the gap with the most affluent groups.

Thus, this paper is organized as follows. First, we conduct a brief discussion of the main concepts and works linked to the study. Then, we present the methodology used in the work, and subsequently the general context of the situation in Santiago de Chile. In the final section, we offer evidence that shows the displacement of the center of services in Santiago and its impact on travel time, ending with a discussion showing these findings.

## 2. Urban Structure and Urban Transport Justice

When we talk about a process of urban transformation, such as an "extended activity center" (EAC) that spans from its Central Business District (CBD), it is important to start considering urban structure theories and the peculiarities of Latin American cities. Cities' structures are geographical expressions of an economic system articulated within its space. Castells [5] addresses this topic by stating that "the urban space is structured; that is, it is not organized randomly, and the social processes that refer to it express the determinants of each period of social organization." Under this logic, the city is ordered according to the characteristics and the relevance of economic activities, assigning land uses to the role they have in the productive structure of the city.

The correlation between the structure of a city and its economic activity has been addressed through different approaches. The theoretical formulations on urban structures proposed by the School of Social Ecology of Chicago marked the beginning of a long discussion. Burges [6], Hoyt [7], and Harris and Ullman [8] proposed models of a concentric city, sectors, and multiple nuclei, respectively, receiving much criticism. They attempted to explain urban development in socioeconomic conditions characterized by the degree of social homogeneity, the economic base, property, the transportation system, and the function of the central urban nucleus [5]. In this evolution, the services sector becomes more prominent in attracting investment, including a cooperative component due to the need for more space driven by economic growth [9]. This triggers more demand for urban space and the creation of business and commerce districts, which often satisfy certain location patterns that tend to expand the traditional center of activities [10]. According to Janoschka [3], after the post-war period, attempts to model different cities were intensified. Models for Arab and European cities were proposed under both western and socialist regimes. The first attempts to theoretically model a Latin American city can be found in the study by Bähr [11], and in the specific case of the Chilean cities in the attempt by Borsdorf [12]. However, since then, Latin American cities have been transformed as a result of the development of the region's countries and the emergence of new urban artifacts that have changed the structure and landscape of its metropolises.

According to Janoschka [3], these transformations cannot be considered as the mere continuation or intensification of the tendencies that dominated the planning and urban construction until the eighties. Instead, they constitute an evolutionary drift, requiring a new abstraction to understand its configuration. Thus, new theories have attempted to understand the changes present in this new structure. The match of Santiago with the urban evolution model proposed by Griffin and Ford [13] and Ford [14] for Latin American cities is quite remarkable. The model proposed by these papers is presented in Figure 1a. Adapting this diagram to Santiago requires rotating it and enlarging few areas, as is shown in Figure 1b.

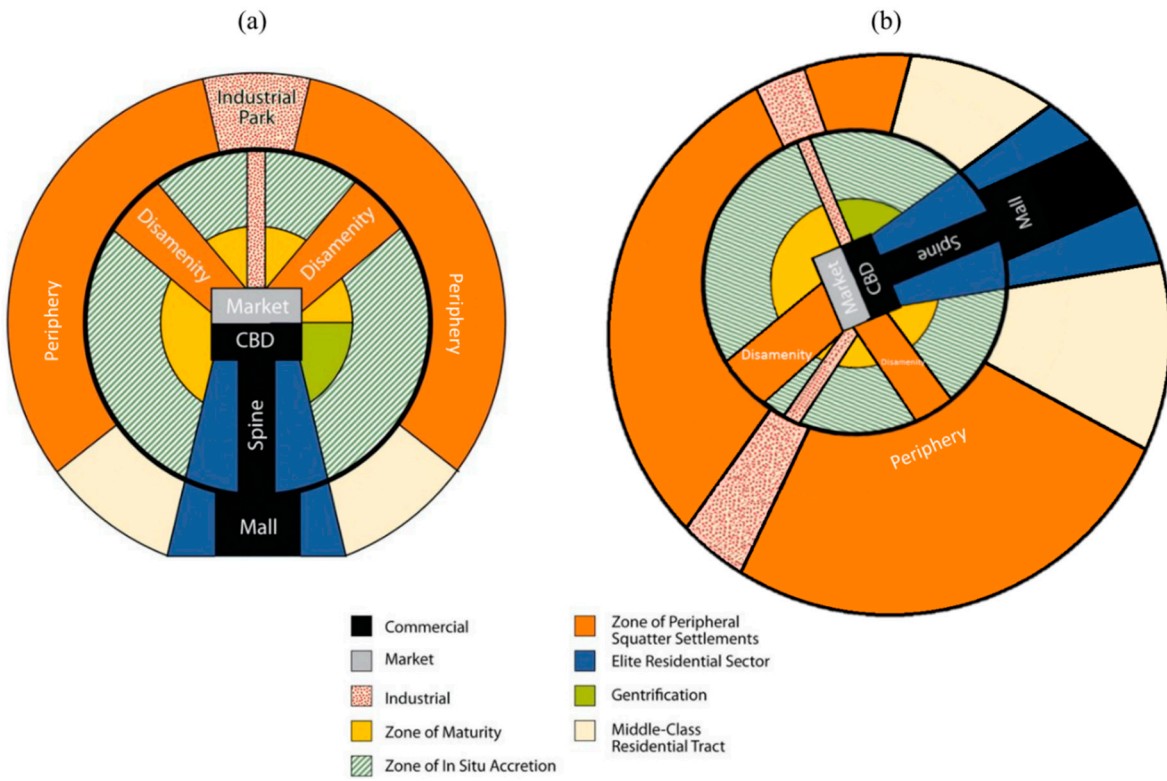

**Figure 1.** (**a**) Urban development model proposed by Griffin and Ford (1980) and Ford (1996) for Latin American Cities (**b**) An adaptation of this model to the evolution of Santiago de Chile.

More recently, "the new model of a Latin American city" [1–3], which is characterized by so-called "territorial fragmentation," was proposed and is the basic principle that determines the dispersion of infrastructure and urban functions. Other authors have made different observations, especially regarding the relationship between the functioning of the land market and the production and reproduction of its use. The model generates a particular urban structure characteristic that Abramo [15] has called a "com-fusa," referring to this double process of expansion and compaction that cities currently experience. Practically, all these urban models include a "high district" or "high-income sector" as a distinguishable aspect of a Latin American city; characterizing its evolution is one of the objectives of this work.

The main contributions of this paper focus on characterizing 25 years of this evolution process in Santiago, focusing on the main axis (represented as Spine in the model) connecting the central business district (CBD) with a high-income residential sector. This process is characterized by analyzing the investment of new infrastructure developments towards the northeast and highlighting the very different travel time consequences for some high-income and low-income groups in Santiago. Regarding the evolution of city centers, Greene and Soler [16] identified several different patterns: linear, nuclear, and circular. They also considered cases where central or pericentral areas become

economically obsolescent. Some relatively small sectors of Santiago have experienced a significant gentrification process. Truffello and Hidalgo [17] also studied the distribution patterns of commerce through the analysis of service density data and land prices, using parametric models. They concluded that although Santiago shows a monocentric patter, it shows some trends regarding services towards a polycentric urban structure. Vásquez and Fuentes [18] studied the evolution of office locations in Santiago, identifying a linear growth pattern towards the northeast. However, its growth was not geographically continuous. Instead, more distant areas were filled with new offices, while the areas in between were filled over several years. This process has been transforming the structure of the city and generating an extension of the metropolitan center. Other authors such as Contreras [19] referred to downtown Santiago as a kaleidoscope due to the social diversity that can be observed. This diversity can be considered as an exception in a city that is composed of very homogeneous neighborhoods, such as the sector that we characterized as the extended activity center. All these studies suggest that the evolution of Santiago has been consistent with the urban structure model by Griffin and Ford. The presence of an extended activity center as is highlighted in the model is featured in this work.

For the case of Santiago, we show that the consolidation of the high-income sector under this urban structure intensifies the problems of accessibility and social exclusion, because those who have less resources are faced with longer trips to reach the city's main services. Most members of these low-income groups are captive users of public transport, specifically the bus system. They mostly lack the possibility of travelling by car, which almost always achieves a speed higher than that offered by the public transport system (especially with the construction of urban highways). Also, from the point of view of the resources needed by the public transport system, the new end locations demand longer trips, implying a more costly system in terms of capital investment and operation. Furthermore, the unfairness in concentrating the locations of this group of highly intensive transit users in certain periphery areas triggers an uneven use of public transport infrastructure, transit is heavily focused in one direction during peak hours by these travelers, while the other direction shows noticeably less activity.

Some authors have focused their work on access to urban transport. Gössling [20] states that contemporary transport systems are characterized by injustice. He conceptualizes "urban transport justice", proposing three dimensions where injustices are apparent: exposure to traffic risks and pollutants; distribution of space; and valuation of transport time. Similarly, Martens [21] developed a new paradigm for transportation planning based on principles of justice. The author observes that for the last fifty years, the focus of transportation planning and policy has been on the performance of the transport system and ways to improve it, without much attention being paid to the persons actually using—or failing to use—that transport system. In this context, the spatial distribution of activities and the available transportation systems become key, since these delineate the maximum set of opportunities or maximum level of accessibility persons can obtain. These opportunities are captured by the concept of accessibility, which is important in transport planning and has been paid increasing attention in the literature in recent decades.

Thus, the relocation of less affluent people in Santiago into its periphery has affected accessibility due to longer trip distances, travel times, waiting times, and higher discomfort. Long trips also increase the cost of the transport system. Leonor et al. [22] examined "how far the restricted mobility and activity patterns of citizens in these low-income communities influence or interact with their quality of life outcomes in terms of their wealth, health, and wellbeing." However, in this paper we argue that these users are suffering a new problem. During the last few decades, Santiago has systematically expanded the location of its main activities from the historic downtown center towards the northeastern sector, which has progressively impacted these users' living standards.

This evolution of the city has further deteriorated the travel experience of the residents living in the southern and western low-income peripheries of Santiago, highlighting the impact of their longer and more expensive trips on the public transport system when compared to other areas in Santiago, as has also been evidenced by the low-income peripheries in Colombian cities [23,24]. As a counterpart,

the more affluent sectors located mainly in the northeastern sector of Santiago benefit from this urban drift, enjoying work locations closer to their homes, reducing their travel times.

## 3. Methodology

To calculate all the indicators, official databases provided by public institutions were used. The formulas are detailed below.

### 3.1. Service Construction Growth Indicators

The National Revenue Service (SII) database registers the built surface area, measured in square meters, for each of Santiago's addresses since 1990 and disaggregates it into different purposes, with services being the most prevalent category for typical business offices found in a downtown area. Thus, to characterize the evolution of the new investments in service infrastructure, the growth of the surface area devoted to services during 1995, 2000, 2005, 2009, and 2015 with respect to 1990 is computed. Thus, for any zone $i$, the construction of infrastructure from type $Z$ in year $X$ since 1990 would be:

$$construction\ Z_i^X = surface\ Z_i^X - surface\ Z_i^{1990} \tag{1}$$

This indicator presents the total surface area in each category in each zone during a certain period. However, to associate such an indicator with the evolution of the attractiveness of different zones, it is necessary to normalize it using zone size. Thus, a second indicator is proposed, denoted as the percentage change in surface infrastructure devoted to services in each zone, shown as:

$$percentage\ Z_i^X = \frac{surface\ Z_i^X - surface\ Z_i^{1990}}{surface\ _i} \tag{2}$$

Both indicators are used here to analyze the dynamics of activity locations in Santiago.

### 3.2. Aggregated Interzone Travel Time Indicators

Travel time was used to compare connectivity from different locations to the extended activity center, the area spanning from downtown towards the northeast, in which most city jobs are concentrated and which attracts the most trips during the morning rush hour. To estimate this travel time from location $i$ to zone $j$ in a given year, information from Santiago's Bip! travel cards was used. Although travelers must only tap-in to enter each vehicle, the transport authority has managed to estimate the origin, destination, and travel time for most trips [25,26]. This data source also assigns an expansion factor to recognize that out of all trips originating from a given stop, the destination can be identified for only a fraction of them. There, the expansion factor amplifies the trips from each stop, assuming that the destination distribution is the same for fully characterized and uncharacterized trips. Thus, the average travel time is computed as follows:

$$TT_{i,j} = \frac{\sum_{n \in N(i,j)} TT_n \cdot ExpansionFactor_n}{\sum_{n \in N(i,j)} ExpansionFactor_n} \tag{3}$$

The set $N(i,j)$ corresponds to all trips detected in the database whose origin is within zone $i$ and whose destination is within zone $j$. This indicator and the number of travelers between the respective pair of zones is used to build a cumulative distribution function of the travel times of real trips from location $i$ to any zone comprising each of the five sectors. These cumulative distribution curves are used to compare the percentage of the trips from different locations that can reach a given sector within a travel time of $T$.

## 4. City Context and Evolution of Infrastructure Investments with Service Purpose

Santiago de Chile has expanded significantly in recent years. In 1991, the total urban area was 50,181 hectares, with a population of 5.1 million people [27], In 2012, these values grew to 72,913 hectares and 6.9 million inhabitants [28]—increases of 45% and 34%, respectively, decreasing population density from 103 inhabitants/hectare to 95 inhabitants/hectare. At the same time, there was an increase in residential production in the inner city, especially in the commune of Santiago, which had one of the greatest growth rates of inhabitants during the last 10 years. Presently, the city has different growth trends, combining expansion and verticalization [29]. Nevertheless, this growth has not been uniform across the whole metropolitan area, with social housing policies being the main reason behind this trend.

For decades, social housing has been built in peripheral sectors, mainly to the south and west of the city (36% of the total housing built between 1978 and 1995 was in three southern communes, out of a total of 34 in the whole city). Families previously living in informal settlements relatively close to the urban center of activities were relocated into new peripheral social housing projects. In contrast, between 1979 and 1992 in the northeastern area of the city, no social housing projects were developed [30]. Instead, this part of the city concentrated on growth of services and jobs, generating a shift away from the traditional and geographical urban center where these opportunities used to be based. For example, in the Las Condes commune, located in the northeastern area of the city, 3.1 million square meters of buildings for the services sector were built during this period. In contrast, in Lo Espejo (a low-income district, with an area close to one tenth of the Las Condes area but closer to downtown), only 35,000 square meters were built. It is quite clear that social housing policies contributed to the current social inequality of the city. This adds complexity to social policies, considering that segregation produces an uneven access to services and to the labor market [31].

Regarding jobs in Santiago, according to the 2002 Census [32], half of them are offered in three communes: Santiago, Providencia, and Las Condes (Figure 2), where only 10% of the population of the metropolitan region lives. Close to 80% of these three communes' residents don't need to move beyond their commune to go to work [31]. Others employed in this region must travel from other communes for work. This generates a great inflow of trips to these communes during the morning peak periods [33,34].

In this section, we show how the central area of Santiago, which attracts most of the city's activity, has experienced a systematic evolution towards this northeast area, in which most affluent people live. Figure 2 shows a map of Santiago, highlighting its northeastern high-income area. The area has been divided into the five sectors that will be used to analyze this phenomenon. The map also displays the geographical distribution of the main landmarks we will use in this paper.

This process has been accompanied by almost completely excluding the low-income population from this area, leading to the consolidation of an exclusively high-income area surrounding the Alameda–Providencia–Apoquindo–Las Condes (APAL) corridor. The exclusion of the low-income population from this area is symbolized by the eradication of all its informal settlements between 1979 and 1985 (see [35] for a detailed description). Figure 2 provides a metropolitan perspective of the main displacement processes, indicating the original locations of the informal settlements (in red) and the final establishment of the affected families in social housing neighborhoods (in blue). The figure also shows that the affected families have been located farther from the activity center than they were before. It can be argued that the eradication of the settlements in this area was one factor in the consolidation of the high-income sector seen today. However, the causality behind this trend remains unsolved.

The steady, uninterrupted expansion of the city center towards the northeast has not been a concern for citizens or authorities in Santiago, even though it heightens the city's inequality and negatively affects quality of life and access to opportunities.

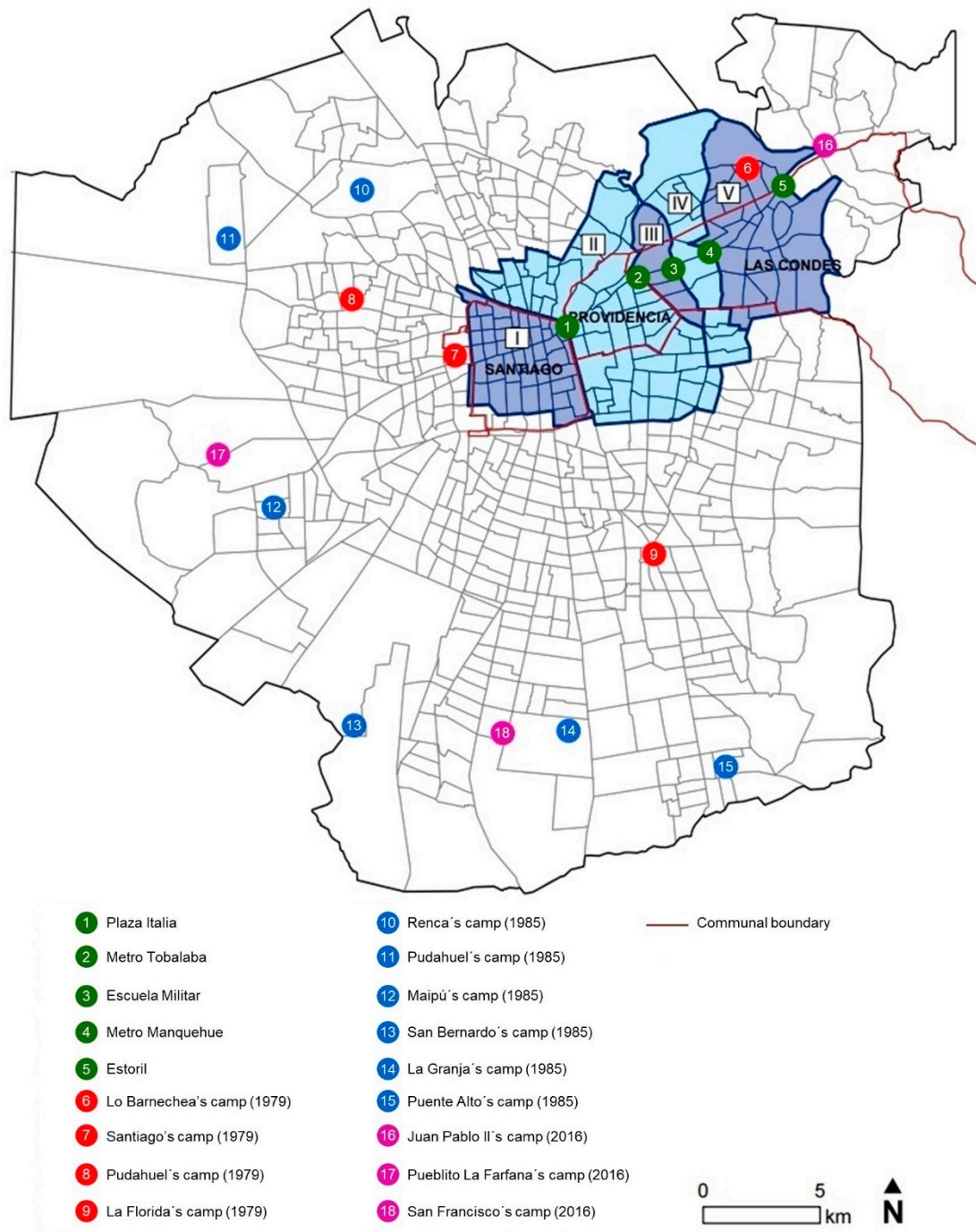

**Figure 2.** Main landmarks and informal settlements in Santiago (Source: created by authors based on [35,36]).

This section intends to characterize part of this expansion by quantifying the built area used for services (public administration, offices, and health services), which constitutes a strong generator of job opportunities, and therefore, is a driver of commuter trips. In this 25-year analysis, the evolution of this type of activity towards the northeast will be shown. To visualize this expansion, QGIS open source software is used [37].

To characterize the evolution, we define a study area we have denoted the "extended activity center" (EAC) area. This area includes the historic business district and all the northeastern surroundings of the APAL corridor. The implicit hypothesis is that this area not only contains most of the new

infrastructure investments for service purposes, but also that these new investments are increasingly located towards the northeast.

The EAC is divided into five sectors, whose limits correspond to five geographical landmarks along the corridor, selected due to their importance to the city. As shown in Figure 2, the selected landmarks correspond to Plaza Italia, Metro Tobalaba, Escuela Militar, Metro Manquehue, and Estoril. They are the northeastern limits of the five sectors, which we will name sectors I, II, III, IV, and V, respectively.

Analyzing sectors I–V (which cover 12.4% of the Greater Santiago area) with Equation (1), we conclude that this small portion of the city covers about two-thirds of the total surface area being added citywide for service purposes. However, if we look to the areas closer to the business center, they do not capture a constant fraction of the total surface area being added.

To better understand this phenomenon, Figure 3 displays how this growth of the surface area built for service purposes in the EAC is distributed between the five sectors for the same five temporal periods (periods of five years).

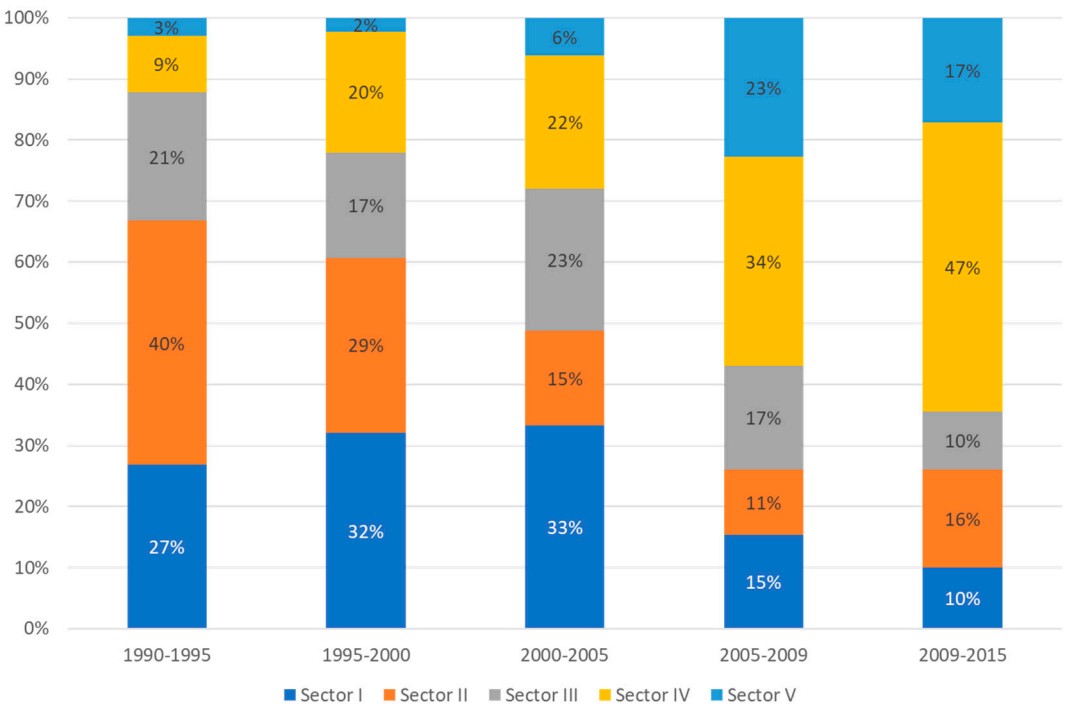

**Figure 3.** Percentage increase of surface area devoted to services in each sector (Source: created by authors based on data from the Chilean Internal Revenue Service (SII database).

The evolution presented in Figure 3 shows that the investment for service purposes has increasingly been focused in areas further away from the city center. While sectors I and II saw a clearly declining trend during these 25 years, the decline of sector III was more gradual. The sector that shows the strongest growth is sector IV, while sector V is beginning to show the trend experienced by sector IV a decade before. Thus, based on this graph, it is reasonable to infer that sector V will become the target of new service investments in the decade to come.

Thus, this analysis confirms that the construction of services is heavily concentrated in the high-income northeastern area and that it is continuously evolving farther away from the business center and towards the northeast of the city along the APAL corridor. This result is consistent with other studies on the evolution and change in the localization patterns of offices and services in Santiago [18].

## 5. Impacts in the Transport System

The evolution of the new investments with service purposes has had an impact on the number of commuter trips towards this area. To identify this impact, the most detailed information available is the origin–destination matrix for public transport users, developed following the methodology presented in [25,26]. The matrix indicates (sometimes based on predictions) the station of origin and destination for each paid trip in the system. This matrix is built based on data provided by automated fare collection and automated vehicle location systems. Unfortunately, this information has been built only for the period between 2011 and 2015, so the evolution of trips between 1990 and 2010 cannot be identified. Based on this information, we analyzed the distribution per sector of the trips reaching the EAC during the morning rush hour in 2011 and 2015. Despite the short time interval, a clear trend is observed, in which the distribution increases for sectors III, IV, and V, which are farther from downtown, while it drops for sectors I and II.

Based on the same information, we evaluated the percentage change of trips being attracted by each of these sectors, observing that the number of trips heading towards sector III grew by more than 20% in just four years. Sectors IV and V also saw an increment in trips of five percent and 13%, respectively, while the trips heading towards sectors I and II declined by six percent and 10%, respectively.

Thus, this data confirms that the evolution of the new investments in the extended activity center towards the northeast has been accompanied by a displacement of travel destinations towards the same sector.

The previously described evolution of the center of activities in Santiago has important consequences for its public transport system. Since the center of activities is the preferred destination for commuting trips in Santiago, and as most public transport trips approach the downtown from the southern and western peripheries (see Figure 2), the length of these trips grows. Thus, it is reasonable to expect that the cost of providing public transport services and infrastructure will also increase. In June 2017, a new metro line was announced, aiming to provide a fast connection between the northwestern area of Santiago and the northeastern border of sector V (Estoril). This example shows how the investments in the transportation system of Santiago follow the land use evolution of the city. Since the city lacks a metropolitan land use plan, land use is determined largely by private initiatives. Thus, it is reasonable to ask if the evolution depicted in the previous section of this paper is the most beneficial for the city as a whole.

This evolution has had a heterogeneous impact on the travel times of the people in Santiago—while those located in the northeastern area presumably benefit from this process, those living in the southern and western areas of Santiago suffer from it. Analyzing the impact on travel times at the city level would require data for trips starting from different origins in the city and for different modes over several years. Instead, this article focuses its analysis on the impact of the EAC evolution on one specific group: the residents of informal settlements that were displaced from somewhat central locations in Santiago towards its periphery between 1979 and 1985, as is detailed in the next section.

*Travel Time for Dwellers of Informal Settlement Eradicated between 1979–1985*

The evidence of other countries shows us that "greater and more equal access to economic opportunities, local services, and facilities will encourage stronger social support if opportunities expand within, rather than without, cities" [38]. In Santiago, one of the main housing policies since 1979 has been the formalization of informal settlements that were in the eastern sector of Santiago, moving them to other communes. In fact, a cadaster prepared by the Ministry of Housing and Urban Planning of the Metropolitan Region detected more than 294 irregular settlements that served as homes for more than 44,000 families and almost 223,000 people. Hidalgo [39] states that between 1979 and 1986 alone, more than 60% of the families that inhabited camps were evicted and accessed new social housing on the outskirts of the city, in communes that did not have infrastructure or means to receive

these new inhabitants, resulting in a city with a polarized structure. This was key to the phenomenon described in the previous section regarding the consolidation of conditions for EAC.

In this context, and in this specific case, to analyze how this eradication of settlements affected the travel times of the displaced families, this analysis is focused on the 4 informal settlements involving the largest number of displaced families; that is, Santiago, La Florida, Las Condes, and Pudahuel (see Figure 2).

Rigorously comparing how the resettlements affected the displaced families' travel times would require a thorough analysis of their trip destinations before and after the eradication. It is possible that once their location changed, many of their activities (such as school or work) changed too. Three decades later, it is even more likely that the destination distribution of these families is different than it would have been if they had remained in the same location. Therefore, only a preliminary assessment of the impact of these relocations on the travel times of these families is provided here. To do this, the travel times from each of these locations to the 5 previously mentioned sectors are compared. For this comparison, the destinations of the trips emanating from each new settlement are assumed to be those observed in the detailed origin–destination matrix for public transport users developed for a week of April 2015.

When the travel times of trips originating from the former location of the informal settlement of Las Condes (Figure 4) to the EAC are analyzed, it can be seen that 90% of these take less than 40 min. However, only 60% of the trips from the other three origins (of these three, only Pudahuel is displayed in the Figure) take less than 60 minutes to reach the EAC. The average travel time from the Las Condes settlement to the EAC is 26.6 minutes, while from the settlements in San Bernardo, Pudahuel, and La Granja, the average travel time is 51, 59.5, and 56.3 minutes, respectively.

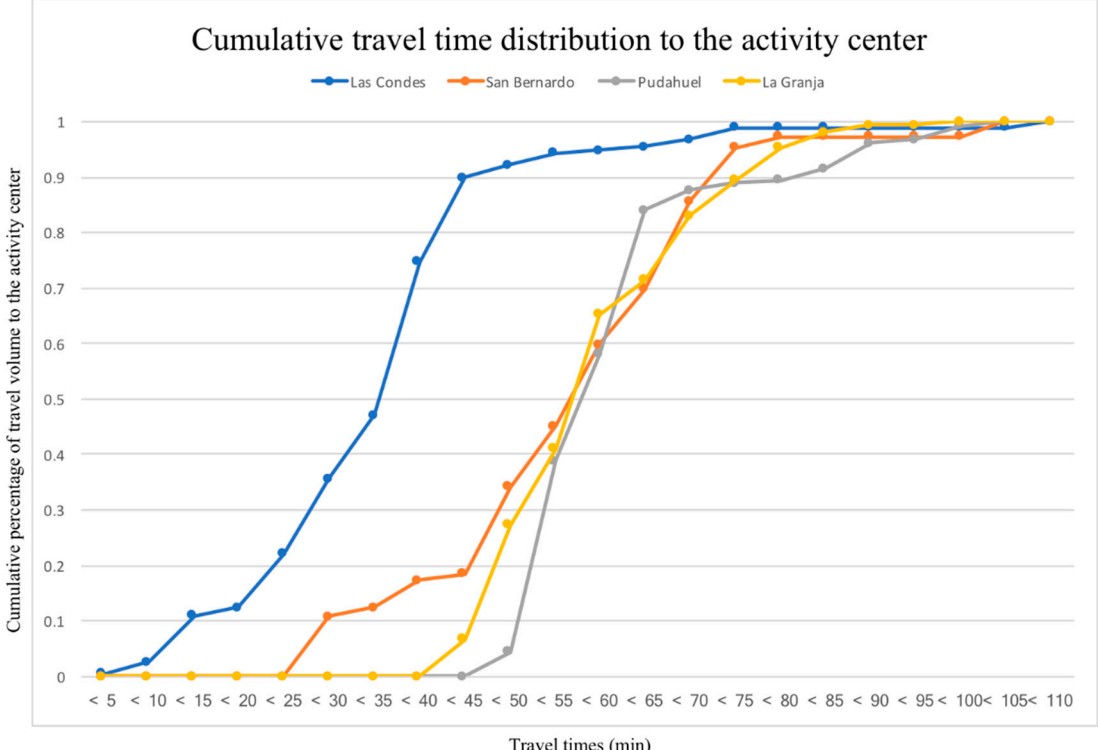

**Figure 4.** Cumulative distribution of travel times to the extended activity center (EAC) from: (i) the location of the original Las Condes slum; (ii) the locations in San Bernardo, Pudahuel, and La Granja, where these families were given social housing (Source: created by authors based on Bip! Data for 2015 [25,26]).

The figure presents the trip time distribution from the three most important destinations for the families that were displaced from the Las Condes informal settlement: San Bernardo, Pudahuel, and La Granja. Thus, Figure 4 quantifies the magnitude of the deterioration in travel times to the EAC for families who were displaced from their informal settlements in Las Condes. Although not shown here, the case of the families displaced from the eradicated informal settlements in the commune of Santiago is even more noticeable: 70% of the trips towards destinations in the EAC have a travel time lower than 25 min, while no trips from the eradication destinations can reach the EAC in that period of time. The average travel time from the Santiago settlement to the EAC is 10.2 min, around 20% of the average travel time needed from the San Bernardo and La Granja settlements, where most of these families were displaced. When repeating this process for eradicated informal settlements in more peripheral areas, such as Pudahuel and La Florida, this trend exists, but to a lesser extent.

With respect to the current situation in Chile and the metropolitan region, according to the 2016 Compilation of Informal Settlements [36], there are 38,770 families that still live in a total of 660 informal settlements throughout Chile. Within the Metropolitan Region of Santiago, which includes Greater Santiago and some rural areas, there are 81 informal settlements containing a total of 4337 families, corresponding to approximately 13,011 people. The informal settlements with the largest number of families in the capital are Juan Pablo II (700 families), San Francisco (300 families), and Pueblito La Farfana (130 families).

Following the same methodology explained previously, travel time distributions from the three main informal settlements in Santiago were created. The locations of these three informal settlements and their relationships to the activity center are presented in Figure 2. The informal settlement located in Juan Pablo II is significantly better located for reaching the EAC than San Francisco and Pueblito La Farfana. Travelers from the Juan Pablo II slum take 32.7 min on average to reach the EAC, and 86% of these trips reach their destination in less than 45 min. Travel times from the Pueblito La Farfana settlement range between 45 and 55 min, and from San Francisco there is a quite uniform distribution between 40 and 75 min.

Those living in the informal settlement of Juan Pablo II benefit from the EAC's evolution towards the northeastern area of the city, while those from Pueblito La Farfana suffer from this evolution. This fact should be considered if authorities in Santiago are requested to relocate families from the Juan Pablo II informal settlement, since they would lose a privileged position. Authorities should also consider the impact of land use regulations on this evolution; if the EAC were to evolve along both directions of the APAL corridor as a result of adequate regulation, then inhabitants of Pueblito La Farfana would benefit too.

## 6. Conclusions

In this paper, evidence has been provided that proves that Santiago's center of activities has expanded towards the northeastern area of the city, which is coherent with the urban expansion model proposed by Griffin and Ford [13] for Latin American cities. The pace of the concentration of new investments for the services sector in the high-income areas of the city has also been characterized. The data shows that in Santiago, two out of three new surface area units for this purpose are located in this high-income zone, which represents only 12% of the urbanized metropolitan surface of Santiago. The analysis also shows that this evolution has been accompanied by an increasing number of commuter trips headed towards the northeastern area of the city. These trips come from all areas of the city, and most of them are negatively affected by this trend, since it increases the distances to be traveled.

Regarding the informal settlements, this work has shown that the eradication policy that took place between 1979 and 1985 displaced these families away from the area that today is populated by a concentration of high-income residents in Santiago. Indeed, there are currently no informal settlements in the EAC, which may be considered as a driving factor for the development of this high-income sector. The eradication process was detrimental to the residents, since their travel times into the extended

activity center grew once relocated. This analysis should be a valuable input when discussing the impacts of the displacement of low-income residents from well-located informal settlements.

In summary, there is a systematic growth of the EAC towards the most affluent sector, located in the northeast of the city. Also, the eradication of informal settlements affected the quality of life of these people, and arguably contributed to the observed evolution of the EAC.

The results of this research provide evidence regarding the need for integrated public policies regulating land use, social housing policies, and transportation. The observed unregulated evolution of the city's structure is causing low-income sectors to face increasing travel times, while the most affluent groups seem to benefit from it. This fact should attract the attention of urban authorities in Latin American cities, several of which have recently faced aggressive protests related to the significant inequality observed in cities. These protests have consistently targeted public transport services.

The EAC evolution pattern also affects the finances of the public transport system, since additional transport capacity are needed (and associated external factors increase, such as accidents and pollution) when the average distance per trip grows. This should draw the attention of city planners and authorities, ensuring that the state has a more active role in governing how the urban structure evolves, monitoring the impact on different communities, especially those that have little choice in where they live and work.

The new transport infrastructure is one of the tools with which the city's evolution could be governed. City authorities can choose to invest in transport infrastructure that will contribute to this evolution, or induce a different pattern by improving the accessibility of well-located but otherwise ignored areas of the city, and in this way foster job market growth in these areas.

The evidence obtained from Santiago is quite consistent with other Latin American cities, suggesting that "fundamental challenges have to be faced, especially those relating to the integration of transport policy, land use planning, and social welfare policies, particularly housing policies" [22].

Finally, it is important to say that the main purpose of this work is to analyze travel times. Therefore, as a future avenue of research it could be considered to both integrate other variables into the study and to replicate this analysis for future years.

**Author Contributions:** This paper was written based on the Master Thesis of G.S.-V. The research topic was proposed by his supervisor, J.C.M. G.S.-V. gathered all the information needed for this work and did all the calculations and analysis presented in the paper. J.C.M. oriented the work identifying research opportunities and analyzing the information that was being obtained. L.F. was a member of the evaluation committee of the Thesis. L.F. suggested the structure that the paper should have and provided the literature background that was key to determine the contributions of this paper to the literature. G.S.-V., J.C.M. and L.F. did the writing of the paper as a team. All authors have read and agreed to the published version of the manuscript.

**Funding:** This research was supported by the Centro de Desarrollo Urbano Sustentable, CEDEUS (Conicyt/Fondap 15110020) and the Bus Rapid Transit Centre of Excellence funded by the Volvo Research and Educational Foundations (VREF).

**Conflicts of Interest:** The authors declare no conflict of interest.

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
