# Peer review of "The Displacement of Santiago de Chile’s Downtown during 1990–2015: Travel Time Effects on Eradicated Population"

_sustainability, doi:10.3390/su12010289_

Round 1
Reviewer 1 Report
This paper studies the relationship between the consolidation of a new central area of Santiago (Chile) and its impact on travel time for inhabitants who lived in informal settlements and were relocated to other sectors of the city.
I find the topic of the paper interesting but the research is not well designed. For instance, the research goal, the methodology and the shown results are not aligned. In addition, some parts need further explanation.
In addition, the authors need to address some issues as follow:
The theoretic background miss references related to the identification of urban central areas. The methodology proposes just one indicator (percentage Z) for this task, but it looks quite simple. In other part of the article it is mentioned that the number of trips to an area could be an indicator of this area centrality. In the results, a table with the values of these indicators for each sector should be shown and I would try to measure if there is any correlation between them.
Regarding the other indicator, the average travel time, I have a doubt: the expansion factor is calculated from the whole set of trips between the origin i and the destination j? If this is so, the denominator is left over, because the sum of all expansion factors should be 1. Isn’t it?
In Figure 1, I can’t find the commnunees Providencia and Las Condes…
Regarding Figure 2, can you specify which data you have used to make it? If it is based on percentage data the chart type used, 100% stacked column chart, it is not suitable because you are calculating the percentage from already a percentage data. A simple stacked column chart, it would be better. I would recommend making a new figure based on a heatmap of the average travel time.
Check the writing, some research keywords used along the paper have no literature background. For instance: service surface, extended activity center… It is important to find the correct terms.
Finally, a miss some references from Sustainability journal, and from some citations such as the National Revenue service (SII).
Reviewer 2 Report
Manuscript “The displacement of activities in Santiago de Chile in 1990-2015: impact on travel times in the city and its informal settlements” is a case study.
The article structure is incorrect. The Sustainability Journal is an international journal and a broader research context should be provided. Research has no significant scientific impact. In addition, they do not formulate new methodology. The results are local and it is difficult to determine their usefulness for the analysis in other cities.
Detailed recommendations for the authors are presented in the further part of the review.
Comments for authors
I suggest minor corrections to the language, especially comas.
Subject
It corresponds to the content.
Keywords
Part of the words chosen correctly, but the part repeats the title.
Abstract
The summary partly reflects the content of the article. It contains the main results without description of research methods. It should be added. It does not contain the purpose of the work, which is a major mistake. The aim of the manuscript was determined indirectly. The abstract should be edited.
Introduction
The content of this section is not an introduction to the issues. This is a description of the research area, which should be moved to the Materials and methods section (subsection 'Description of the research area'). Introduction should be on the top of the manuscript, but in this case the introduction is the characteristics of the research area. Section 3 is a typical introduction (although in the final part it describes the conditions of the research area) and should be moved to the Introduction section. Lines 51-60 should stay in the Introduction section – they contain the purpose of the work and the research plan.
2. The urban structure and dynamics of location of travel attractors.
This section contains elements of an introduction. However, it is poor and does not refer to very rich global achievements. The literature review should be extended.
Lines 91-96 refer to the purpose of the work and can be connected with lines 51-60. Lines 117-130 are a further description of the city – but placed in another part. This makes a big mess.
Methodology
3.1. Service construction growth indicators
Correct description of the used indicators. They are simple, but correctly illustrate the dynamics of the analyzed phenomenon.
3.2. Aggregated interzone travel time indicators
Correct measurement of travel time.
City context and evolution of infrastructure investments with service purpose
This part also contains the characteristics of the research area. This should be combined with other parts, previously recommended in the review.
Figure 1 is difficult to read. Descriptions on the right and left should be placed under the figure (as a legend). This can increase the size of the map.
Impacts in the transport system
In this section, the results intertwin with the discussion. The best layout is to separate these elements. However, in this case it makes it easier to analyze the described relationships. I allow this arrangement of results and discussions.
5.1. Travel time of informal settlement dwellers eradicated between 1979–1985
Correct analysis of the results supported by examples from other references.
Main findings (incorrect section numbering)
The title of the final section should be: "Conclusions". First, it repeats the insights from the discussion without adding new significant information. Secondly, general arrangements are made. There is no evidence of the research impact on the possibility of development or recovery strategies. Recommendations are needed for local authorities in other cities – including cities from outside South America.
Literature - items about cities in South America prevail. For this reason, a global context is missing.
Reviewer 3 Report
This paper characterizes the evolution of urban structure in Santiago de Chile over a 25 year period by constructing and descriptively analyzing indicators of newly-built surface area and travel time for residents in different parts of the city. The analysis focuses on the role of displacement of informal settlements in facilitating the movement in the city center towards the northeast, and the effects on travel time among those who were displaced.
The paper does a good job of illustrating the movement of services out towards the northeast over the full period under discussion. There is some supporting evidence from changes in rush hour bus commuting patterns, but only for the 2011 to 2015 period.
I found the analysis to be persuasive for the most part. The main findings do go a bit too far in implying net negative welfare effects among those who were relocated: “The eradication process was detrimental to the residents since their travel times towards the extended activity center grew once [they were] relocated.” Yes, that is reasonable if the only effect of moving these people was on their travel times or if travel times were all that mattered to them, but we are not told anything about other effects. For example, did their housing quality improve? How about availability of employment, security, other aspects of environment? It is not necessary for the paper to do a full welfare evaluation of the displaced people, but if it does not it should explicitly limit its conclusions to the impact on peoples’ travel times.
More broadly, it would be useful to include a brief passage in the main findings on limitations of the work (mainly to do with data, presumably) and future avenues for improving analysis in this area.
Specific comments
There is inconsistent use of the thousands separator in numbers: sometimes “.” is used and sometime “,”. In lines 26-27, the wording could be improved in the passage “…especially in the commune of Santiago, which had one of the greatest growth of inhabitants…”. In line 33, there is no need for a comma in “peripheral, social housing projects”. On line 43, “market” should be “markets” In line 97-98, I don’t think you need the “the” in “…characterized by the so-called…” In line 145-146, where you say “it should be controlled for its size” you probably mean something more like “one needs to normalize it using zone size”. In line 223, “temporary cuts” should probably be “temporal cuts”. In line 305, “most important destination” should be “most important destinations”. In line 311, instead of “lapse” I think the authors mean “period of time”.
Reviewer 4 Report
Review
Article details:
Title: The displacement of activities in Santiago de Chile in 1990-2015: impact on travel times in the city and its 3 informal settlements
Article Type: Full Length Article
Keywords: Metropolitan CBD; evolution of travel times; eradication; informal settlements
Ethics
There are no ethical issues/problems for this article
Originality
This article investigates the city structure of Santiago de Chile and how that effects travel time of people living in less affluent areas in the city. I think this is a very interesting topic worth of investigation and shows originality. Thus, I find this article very original.
Structure
Overall the structure of the article is very good.
Language
The language is very good. Although some minor typos should be corrected.
Abstract
Very well written and informative. No further comments.
Title
Very good and interesting. No further comments.
The article
I find the analysis and the findings very interesting. However, I wonder how they relate to the contemporary scientific literature on transport and spatial justice. There are no references in the article to the leading researchers in this field like Karen Lucas, Karel Martens, Stefan Gössling or Mimi Sheller. Mimi Sheller should be especially interesting for this article, since she has focused in her research on Latin America. This article lacks theoretical perspectives related to the research mentioned above. In my opinion this lack should be dealt with prior to publication. The inclusion of those perspectives should add a much deeper understanding of the findings, which improves the scientific value of the study. Also, the perspective on sustainable transport could be included here in order to make the accessibility analysis connectable to current trends in sustainable transport. Please find below suggested references to be included in this article.
My recommendation is the following:
I think this paper is very interesting and has great potential. However, the lack of theoretical perspectives related to transport justice has to be dealt with in order to deepen the understanding of the findings. This, nonetheless, should not be to difficult to achieve.
Reference lists
Gössling, S. (2016) Urban transport justice, Journal of Transport Geography, Vol. 54, pp. 1-9
Koglin, T. (2017) Urban mobilities and materialities – A critical reflection of a “sustainable” development project, Applied Mobilities, Vol. 2(1), pp. 32-49
Koglin, T. and Rye, T. (2014) The marginalisation of bicycling in Modernist urban transport planning, Journal of Transport and Health, Vol. 1(4), pp. 214-222
Lucas, K. (ed.) (2004) Running on empty: Transport, social exclusion and environmental justice. Policy Press, Bristol
Martens, K. (2017) Transport Justice: Designing Fair Transportation Systems. Routledge, New York
Sheller, M. (2014) Racialized Mobility Transitions in Philadelphia: Connecting Urban Sustainability and Transport Justice, City & Society, Vol. 27(1), pp. 70-91
Sheller, M. (2016) Uneven Mobility Futures: A Foucauldian Approach, Mobilities, Vol. 11(1), pp. 15-31

Round 2
Reviewer 1 Report
Point 1: The mention of references analysing the area devoted to services are scarce.
Point 2: Ok, but you are not saying that in the text (line 216). Please, correct the sentence and explain better the Expansion Factor.
Point 4: Something is wrong in Figure 3, the percentages appearing in the columns do not match those on the axis…It seems that you have built the 100% stacked column chart using percentage growth of surface…
Point 5: If you search in Google Scholar “Extended Activity Center” you get just one result and if you search in Google, just 8 results. Results quite poor to be a keyword…
Reviewer 2 Report
The manuscript "The displacement of activities in Santiago de Chile in 1990-2015: impact on travel times in the city and its informal settlements" in the new version can be accepted. Its structure is correct. The corrections made and the responses to the reviews are satisfactory.
Conclusion from the review – the manuscript can be accepted in present form.
Round 3
Reviewer 1 Report
__________________________